# Complications after Posterior Lumbar Fusion for Degenerative Disc Disease: Sarcopenia and Osteopenia as Independent Risk Factors for Infection and Proximal Junctional Disease

**DOI:** 10.3390/jcm12041387

**Published:** 2023-02-09

**Authors:** Alberto Ruffilli, Marco Manzetti, Francesca Barile, Marco Ialuna, Tosca Cerasoli, Giovanni Viroli, Francesca Salamanna, Deyanira Contartese, Gianluca Giavaresi, Cesare Faldini

**Affiliations:** 11st Orthopaedic and Traumatologic Clinic, IRCCS Istituto Ortopedico Rizzoli, 40136 Bologna, Italy; 2Complex Structure Surgical Sciences and Technologies, IRCCS Istituto Ortopedico Rizzoli, 40136 Bologna, Italy

**Keywords:** sarcopenia, osteopenia, lumbar spine, degenerative disc disease, complications, risk factors, spine surgery

## Abstract

Proximal Junctional Disease (PJD) and Surgical Site Infection (SSI) are among the most common complications following spine surgery. Their risk factors are not fully understood. Among them, sarcopenia and osteopenia have recently been attracting interest. The aim of this study is to evaluate their influence on mechanical or infective complications after lumbar spine fusion. Patients who underwent open posterior lumbar fusion were analyzed. Through preoperative MRI, central sarcopenia and osteopenia were measured with the Psoas Lumbar Vertebral Index (PLVI) and the M-Score, respectively. Patients were stratified by low vs. high PLVI and M-Score and then by postoperative complications. Multivariate analysis for independent risk factors was performed. A total of 392 patients (mean age 62.6 years, mean follow up 42.4 months) were included. Multivariate linear regression identified comorbidity Index (*p* = 0.006), and dural tear (*p* = 0.016) as independent risk factors for SSI, and age (*p* = 0.014) and diabetes (*p* = 0.43) for PJD. Low M-score and PLVI were not correlated to a higher complications rate. Age, comorbidity index, diabetes, dural tear and length of stay are independent risk factors for infection and/or proximal junctional disease in patients who undergo lumbar arthrodesis for degenerative disc disease, while central sarcopenia and osteopenia (as measured by PLVI and M-score) are not.

## 1. Introduction

### Surgical Site Infection (SSI) and Proximal Junctional Disease (PJD) Are the Most Frequent Complications after Lumbar Spine Surgery [1,2,3,4,5,6,7,8]

Proximal Junctional Disease is a spectrum of pathologies ranging from proximal junctional kyphosis (PJK) as a radiologic finding with no clinical relevance to proximal junctional failure (PJF) with instrumentation failure, pain, and neurologic deficit [9,10,11]. 

Surgical Site Infection following spine surgery comprises superficial and deep infections. Superficial spine infections involve the skin and the subcutaneous tissue, while deep ones disseminate under the fascia and muscles all the way down to intervertebral discs and vertebral bones [12].

These complications negatively influence functional outcomes and patients’ satisfaction, and often lead to revision surgery [7,8]. Therefore, preventing them would represent a fundamental achievement for surgeons and patients. However, this is made difficult by the fact that their risk factors are not fully understood. A number of patient-specific (age, body mass index, muscle mass, bone quality) and surgical (overcorrection, sacrum fixation and rigidity of the construct for PJD, revision surgery and dural tears for SSI) factors are thought to act together and contribute to the junctional disease and surgical site infection [7,10,11,13].

However, with the increasing age of patients undergoing these surgeries, interest is growing towards specific risk factors. Specifically, sarcopenia (defined as “syndrome of progressive and generalized loss of muscle mass and strength”) and osteopenia (defined as decreased vertebral bone density) have attracted much attention, as part of the so-called “fragility syndrome” [14]. They represent patient-specific risk factors [9,10,15], and, in addition to surgical risk factors [16], are thought to predispose to a higher number of infective and mechanical complications in both orthopedic and spine surgery [15,17]. To the best of our knowledge, no studies have investigated all independent risk factors for mechanical and infective complications after lumbar fusion, also focusing on both sarcopenia and osteopenia.

The aim of the present study was to identify independent risk factors for complications, with particular focus on the impact of osteopenia and sarcopenia, in patients who underwent open posterior lumbar decompression and fusion for degenerative disc disease.

## 2. Materials and Methods

### 2.1. Study Sample

After institutional review board approval (CE-AVEC 208/2022/Oss/IOR), a retrospective analysis of all patients who underwent posterior, open lumbar or lumbosacral decompression and fusion for degenerative disc disease between November 2005 and November 2020 in our institution was conducted. Indications for fusion in our institution were as follows: chronic low back pain with clinical signs of instability; radiographically proven dynamic instability; degenerative spondylolisthesis; central stenosis; significant reduction of disc height; facet degeneration and/or subluxation.

Patients with preoperative degenerative scoliosis (coronal deformity >10° Cobb), flat back, preoperative sagittal imbalance (Sagittal Vertical Axis >5 cm) or coronal imbalance (Coronal Vertical Axis > 2.5 cm) were excluded.

Other exclusion criteria were decompression alone, previous spinal surgeries, a diagnosis other than degenerative lumbar spine disease, lack of preoperative MRI or complete medical records of the hospitalization, less than 24 months of follow-up.

### 2.2. Data Collection

Verbal informed consent to participate was obtained from all the patients before beginning the data collection.

The following information was collected for the study: age, gender, smoking history, diabetes, Charlson comorbidity index (CCI), body mass index (BMI), American Society of Anesthesiology (ASA) score. All the operatory registers were analyzed: number of fusion levels, operative time, perioperative complications were included in the database. Length of stay before discharge and postoperative complications were also reported. The diagnosis of surgical site infection (SSI) was based on clinical and radiographic findings, blood tests and/or a documented positive culture obtained at the time of revision or debridement surgery. The diagnosis of proximal junctional disease (proximal junctional kyphosis-PJK-or failure-PJF) was based on full length standing X-rays taken after surgery and at each follow-up visit. PJK was defined as follows: sagittal Cobb angle between the lower endplate of the upper instrumented vertebra (UIV) and the upper endplate of 2 supra-adjacent vertebra ≥10° [18,19,20]. PJF was defined as symptomatic PJK requiring revision surgery, including vertebral fracture of UIV or UIV + 1, fixation failure, subluxation between UIV and UIV + 1, screws pull-out or breakage, and/or disruption of the posterior osteoligamentous complex [21].

Moreover, preoperative MRIs were evaluated and data about sarcopenia and osteopenia were collected. The Psoas to Lumbar Vertebral Index (PLVI) was taken as a measure of central sarcopenia and the M-Score as a measure of bone density. Both are validated scores [5,20]. The PLVI (Figure 1) was calculated measuring the psoas muscle and the L4 body cross sectional areas (CSA) on a single MRI axial cut of L4 pedicles [5], applying the following formula: (Left psoas CSA + Right psoas CSA)/2/L4 CSA.

The M-score (Figure 2) was calculated on the T1W Spin-Echo-sequence, which is the most accurate for the evaluation of bone marrow. In the sagittal section passing through the spinous process of the lumbar vertebrae a region of interest (ROI) (TR = 7, TE = 400–600, slice thickness = 4 mm, fov = 280 mm, matrix = 320 × 320), was placed in the vertebrae from L1 to L4. When the ROI did not follow the above parameters, it was excluded from the M-score measurement. Cortical bone, lumbar plexus, focal lesions, radiological artifacts, were avoided. To evaluate the noise, a ROI was also placed outside the spine: signal-to-noise ratio (SNR_L1–L4_) was calculated dividing the vertebral body intensity by the SD of the noise. The mean and the SD of the population were used, and the M-score was obtained with this calculation: M-Score = (SNR_L1–L4_ − SNR_ref_)/SD_ref_ [22]_._

The measurements were taken independently by two experienced spine surgeons (MI and TC), both blinded to the patient’s names and the average values were recorded.

### 2.3. Statistical Analysis

Parametric testing was carried out to compare continuous variables and normal distribution. Normal distribution was verified using the Shapiro–Wilk test. Homogeneity of the variables was checked with the Levene test. The 2-tailed Student’s *t*-test was applied for non-homoscedastic unpaired groups. The 2-tailed Mann–Whitney U test was used as a nonparametric test for unpaired groups.

To identify independent risk factors for infection and PJD, multivariate linear regression was performed after adjusting for potentially confounding factors such as chronological age, BMI, gender, comorbidity index and ASA score. *p* values < 0.05 were considered significant. All analyses were conducted with the Statistical Package for Social Science (IBM SPSS Statistics for Windows, Version 26.0; IBM Corp., Armonk, NY, USA).

## 3. Results

### 3.1. Demographics

Three hundred ninety-two patients (197 females—50.25% and 195 males—49.75%) were included. Mean age at surgery was 62.6 ± 6.2 (range 31–84) and follow-up was 42.4 months (range 24.4–124.2). Avereage PLVI was 0.76 ± 0.21 (range 0.29–1.62) and mean M-Score −0.11 ± 0.39 (range −0.78–1.74).

Thirty-one patients (31/392, 7.9%) had a postoperative surgical site infection (SSI), after an average time of 30 days after surgery (range 14 to 43). Various microorganisms were identified as responsible for the infections: Methicillin-susceptible *Staphylococcus aureus* in fifteen cases (48.3%), Methicillin-resistant *S. aureus* in seven cases (22.6%), *Escherichia* coli in five cases (16.2%) and *Enterobacter cloacae* in four cases (12.9%). The overall incidence of proximal junctional disease (PJD) was 15 cases among 392 (3.8%): 5 patients had proximal junctional kyphosis (1.27%) and 10 had proximal junctional failure (2.5%). All these patients required revision surgery. No patient developed distal junctional complications. Data about patients’ demographics are summarized in Table 1.

### 3.2. High vs. Low PLVI Patients

A total of 188 patients (47.95%) of our cohort had low PLVI (LPLVI) and 204 (52.05%) had high PLVI. Some of the baseline characteristics were different between the two groups: LPLVIs patients were more frequently older (65.84 ± 7.29 vs. 62.57 ± 10.02, *p* < 0.001), females (115/392 vs. 76/392, *p* < 0.001), with a higher CCI (2.48 ± 1.48 vs. 2.37 ± 1.56, *p* < 0.001) and ASA score (2.02 ± 0.59 vs. 1.98 ± 0.63, *p* < 0.002). Moreover, their length of stay was longer (12.03 ± 12.3 vs. 10.5 ± 11.45, *p* < 0.025). Nevertheless, low PLVI was not related to a higher risk of infection (15/392 vs. 16/392 *p* = 0.73) or PJD (7/392 vs. 8/392, *p* = 0.76).

### 3.3. High vs. Low M-Score Patients

M-score could be calculated in 341/392 patients, because of MRI quality. Among them, 212 (62.1%) had low M-Score and 129 (47.9%) had high M-Score. The two groups were significantly different only in two variables: low M-score patients were more often diabetic (31/212 vs. 5/129, *p* < 0.008) and had higher length of stay (10.5 ± 11.5 vs. 8.4 ± 9.4, *p* < 0.019). However, when compared to the High M-Score group, they were not at higher risk of developing surgical site infection (17/31 vs. 14/31, *p* = 0.57) and did not show lower PLVI (0.72 ± 0.3 vs. 0.75 ± 0.3, *p* = 0.36).

### 3.4. Infectious Status

A total of 31 SSI were recorded in our cohort, with a rate of 7.9%. Infected patients showed a longer length of stay (12.3 ± 10.7 vs. 10.1 ± 8.9, *p* < 0.001), a higher CCI (3.1 ± 1.38 vs. 2.26 ± 1.46, *p* = 0.014) and were more frequently smokers (80/261 vs. 11/31, *p* = 0.038). However, infected patients did not have lower PLVI (0.76 ± 0.12 vs. 0.75 ± 0.6, *p* = 0.6) or M-Score (0.06 ± 1.04 vs. −0.3 ± 0.58, *p* = 0.68) values when compared to non-infected patients.

### 3.5. Mechanical Complications

A total of 15 PJD were identified, with a rate of 3.8%. Univariate analysis identified some differences between PJD and non-PJD patients, including BMI (26.1 ± 5.9 vs. 23.5 ± 3.5, *p* = 0.043), age at surgery (68.84 ± 3.29 vs. 59.57 ± 10.5, *p* = 0.005), diabetes (10/15 vs. 5/15, *p* = 0.018) and PLVI values (0.52 ± 0.5 vs. 0.85 ± 0.7, *p* = 0.038). As for M-Score values, no significant difference was identified (0.16 ± 1.04 vs. −0.3 ± 0.58, *p* = 0.17).

### 3.6. Multivariate Analysis

Multivariate analysis (Table 2) for SSI identified CCI (*p* = 0.006) and length of stay (*p* < 0.001) and intraoperative dural tear (*p* = 0.016) as independent risk factors for surgical site infection.

Considering PJD (Table 3), age at surgery (*p* = 0.014) and diabetes (*p* = 0.043) were confirmed as independent risk factors, while PLVI was not (*p* = 0.35).

## 4. Discussion

Understanding specific risk factors for proximal junctional disease (PJD) and surgical site infection (SSI) following lumbar spine surgery is of critical importance for surgeons and patients [23,24]. In fact, it could help prevent complications, or at least inform the patient correctly and help choose the most appropriate treatment [25,26]. The aim of this study was to assess the impact of osteopenia and sarcopenia on postoperative complications in a cohort of patients who underwent open posterior lumbar decompression and fusion. In our cohort, 3.8% of patients developed PJD and 7.9% experienced postoperative infection.

We had two main results: first, sarcopenia (low PLVI) and osteopenia (low M-score) were not correlated; second, neither of them represented independent risk factors for infection or PJD.

The first finding was unexpected. In fact, the interaction between muscle mass and bone has been demonstrated. In particular, the presence of a low muscle mass is related with low bone density [27,28]: in fact, a reciprocal interaction between bone and muscle through paracrine and endocrine substances has been demonstrated [28] (Figure 3).

A possible explanation for these surprising results can be found in the design of the present study. In fact, only PLVI and M-score were used to define central sarcopenia and osteopenia: they are both validated scores, but since the first is a parameter of volume and the second a measure of density, the comparison between their trends might be not accurate.

As for the second finding, while our results are in contrast with existing literature regarding the correlation between sarcopenia, osteopenia and postoperative outcomes, they are in line with other authors regarding the other independent risk factors identified.

Considering surgical site infection (SSI), sarcopenic patients were in fact found by Bokshan et al. [14] to have a threefold increase in complications rate after thoracolumbar surgery.

Zakaria et al. [29] reported similar findings on 395 patients undergoing posterior lumbar fusion: they found a low psoas area to be related to higher risk of complications. However, infection was not the only one focus of these studies: the authors included any severe complication. Moreover, these authors did not stratify patients for indication or surgical procedure.

Considering proximal junctional disease (PJD), the influence of central sarcopenia and osteopenia has been studied by other authors. Eleswarapu et al. [7] found a significant association between sarcopenia and PJD patients operated for ASD; however, they included all adults (>18 years) without upper age limit: this could create bias due to the inability to distinguish sarcopenia as a pathological entity from the natural loss of muscle mass due to senescence. Kim et al. [25] found a significant association between the thoracolumbar back muscles volume and the incidence of PJK, but they did not perform a multivariate analysis for other confounding risk factors and did not evaluate sarcopenia with the measurement of Psoas muscles at L4 height. As for osteopenia, Elarjani et al. described it as the only risk factor for revision surgery in a cohort of patients with PJK, while Kim et al. [25] found patients with osteoporosis to be twice as likely as others to develop PJD.

A possible explanation for the discrepancy between our results and those of other authors can be sought in the cohort of patients. In fact, our patients were operated for degenerative disc diseases and not for adult spinal deformities (ASD): therefore, they not only underwent relatively short arthrodesis, which has been described as a protective factor [25], but also, they were not exposed to other common risk factors for PJK and PJF; risk factors such as corrective maneuvers, fixation to the pelvis and non-physiological postoperative alignment were not present [20,21,25,30,31]. This difference between our cohort and the others is also demonstrated by our extremely low range of PJD (3.8%).

Regarding the other results, not surprisingly, comorbidity index, dural tear and diabetes were identified as risk factors for SSI and/or PJD. These results are in line with the current literature, where the negative impact of these patient-specific characteristics has been widely analyzed and proven [1,2,3,4,5,6,7,8,9,25,26,27,28].

The results of the present study should be considered in the context of its limitations. First, data collection was subject to the limitations of a retrospective study. Then, only PLVI and M-score were measured to assess central sarcopenia and osteopenia, representing a potential bias. Moreover, the PJD population is relatively small: therefore, the analysis may be underpowered and some non-significant risk factors might have turned out differently if studied in a larger cohort. Last but not least, the lack of a control group who received therapy for sarcopenia and/or osteopenia represents a bias and inevitably weakens the conclusions.

Despite these limitations, this study has a relatively large cohort, a long follow-up and addresses an important topic.

## 5. Conclusions

Our results indicate that comorbidity index, diabetes, dural tear and length of stay are independent risk factors for infection and/or proximal junctional disease in patients who undergo open posterior lumbar decompression and fusion for degenerative disc disease, while central sarcopenia and osteopenia (as measured by PLVI and M-score) are not.

## Figures and Tables

**Figure 1 jcm-12-01387-f001:**
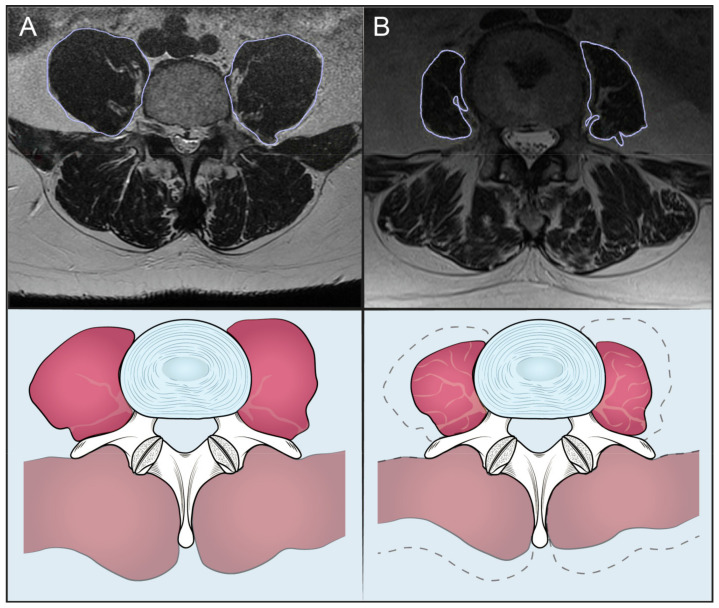
Patients with high (**A**) and low (**B**) PLVI.

**Figure 2 jcm-12-01387-f002:**
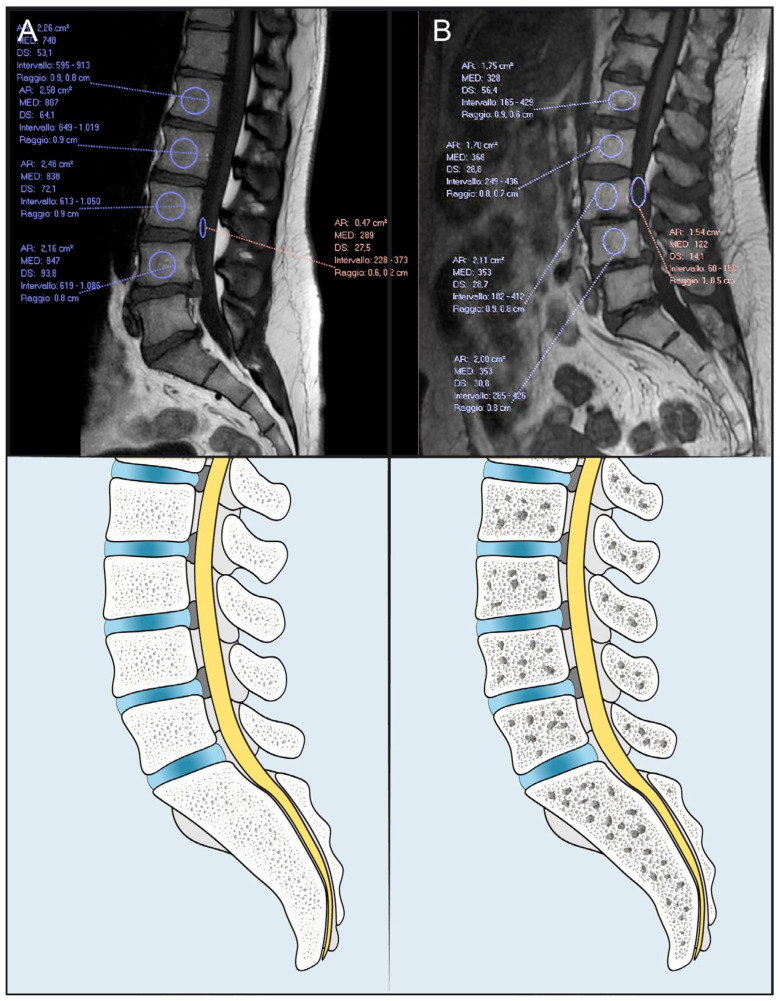
Patients with high (**A**) and low (**B**) M-Score. AR = area, Med = average Hounsfield unit; DS = standard deviation of the Hounsfield unit; Intervallo = interval of Hounsfield unit; Raggio = radius.

**Figure 3 jcm-12-01387-f003:**
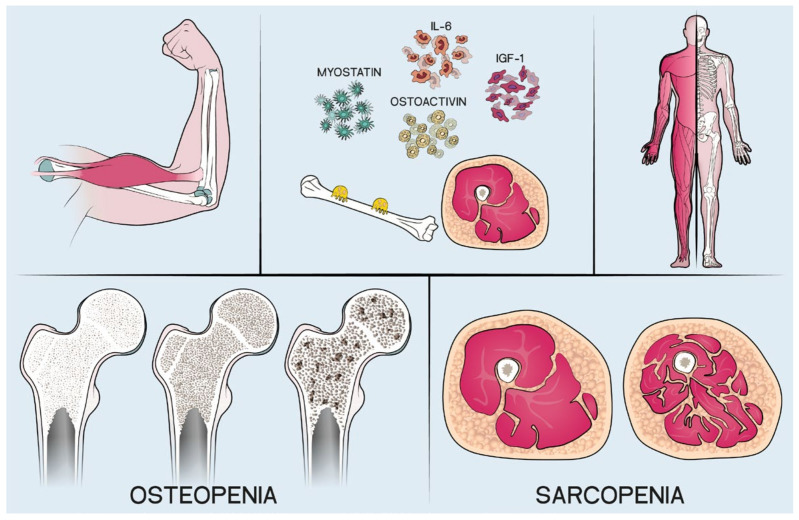
Interactions between bone and muscle.

**Table 1 jcm-12-01387-t001:** Baseline characteristic differences for high vs. low PLVI groups, for high vs. low M-Score groups, for non-SSI vs. SSI groups and for PJD vs. non-PJD groups. * = significant value.

Characteristics	Total	Low M-Score	High M-Score	*p* Value	Low PLVI	High PLVI	*p* Value	Non-SSI	SSI	*p* Value	Non-PJD	PJD	*p* Value
** *n* **	392	212	129		188	204		361	31		377	15	
**Age at surgery *(y. mean.*** ** *± SD)* **	62.6 ± 6.2	62.56 ± 10	62.7 ± 9.9	0.5	65.8 ± 7.29	62.57 ± 10.2	**<0.001 ***	63.6 ± 5.98	65.9 ± 7.96	**0.002 ***	59.57 ± 10.5	68.84 ± 3.29	**0.05 ***
**Gender *(F)***	197	127	69	0.54	115	76	**<0.001 ***	142	18	0.08	197	7	0.6
**Diabetes** **Mellitus (*yes. n)***	36	31	5	**0.08 ***	19	17	0.78	4	24	0.27	34	2	**0.018 ***
**Charlson Comorbidity** **Index *(n. mean.* *± SD)***	2.37 ± 1.6	2.37 ± 3.55	2.39 ± 1.56	0.98	2.48 ± 1.48	2.37 ± 1.56	**<0.001 ***	2.26 ± 1.46	3.1 ± 1.38	**<0.014 ***	2.39 ± 1.56	2.34 ± 1.51	0.88
**American** **Society of** **Anesthesiology Score** ** *(n. mean.* *± SD)* **	2.0 ± 063	1.98 ± 0.62	1.98 ± 0.61	0.07	2.02 ± 0.59	1.98 ± 0.63	**0.002 ***	2.01 ± 0.56	2.31 ± 0.74	0.07	1.98 ± 0.62	1.97 ± 0.6	0.99
**Body Mass** **Index** ** *(n. mean. ± SD)* **	24.5 ± 3.5	24.5 ± 3.6	24.6 ± 3.6	0.07	25.1 ± 34	24.5 ± 3.5	0.99	26.6 ± 3.6	26.5 ± 4.2	0.98	23.5 ± 3.5	26.1 ± 5.0	**0.043 ***
**Tabagism** ** *(yes. n)* **	91	56	35	0.73	45	46	0.6	80	11	**0.038 ***	87	**4**	0.56
**Length of stay *(day. mean.*** ** *± SD)* **	10.5 ± 3.75	10.5 ± 11.5	8.4 ± 9.4	**0.019 ***	12.03 ± 12.3	10.5 ± 11.45	**0.025 ***	10.1 ± 8.9	12.3 ± 10.7	**0.01 ***	10.03 ± 12.3	11.5 ± 11.45	0.44
**Operative time *(min. mean*** ** *± SD)* **	198.3 ± 60	197 ± 60	198.1 ± 60	0.06	185.1 ± 62.5	197.4 ± 57.3	0.25	192 ± 59.1	208 ± 58.1	0.24	182 ± 49.1	202 ± 57.1	0.20
**PLVI** ** *(mean. ± SD)* **	0.76 ± 0.21	0.72 ± 0.3	0.75 ± 0.3	0.36	0.55 ± 0.1	0.88 ± 0.2	**<0.01 ***	0.75 ± 0.6,	0.76 ± 0.12	0.06	0.85 ± 0.7	0.52 ± 0.5	**0.038 ***
**PLVI *(low n)***	188	129	61	0.6				141	12	0.7	181	6	
**M-Score** ** *(mean. ± SD)* **	−0.11 ± 0.39	−0.12 ± 0.39	1.27 ± 8.1	**<0.01 ***	0.06 ± 1.02	−0.06 ± 1	0.36	0.16 ± 1.04	−0.3 ± 0.58	0.17	0.3 ± 0.58	0.16 ± 1.04	0.17
**M-Score *(low n)***	212				129	133	0.6	237	25	0.06	263	12	0.24
**PJD (%)**	3.8%				3.19%	4.4%	**0.53 ***	8	7	0.5			
**Infection (%)**	7.9%	54.83%	45.16%	0.57	7.97%	7.8%	0.76				6.3%	6.25%	0.78
**N. of levels**	2.84 ± 0.96	2.86 ± 0.95	2.79 ± 0.98	0.8	2.87 ± 0.93	2.80 ± 0.98	0.89	2.82 ± 0.96	3.03 ± 0.91	0.23	2.84 ± 0.96	2.80 ± 0.94	0.72

**Table 2 jcm-12-01387-t002:** Multivariate linear regression of risk factors for infection. F = female, PLVI = psoas to lumbar vertebral index. Length of stay, age, comorbidity index, ASA score and M score were independent risk factors for infections. OR = odds ratio (95% confidence interval lower–higher). * = significant value.

	Estimate	SE	T	OR (95% CI)	*p* Value
**Age at surgery**	−0.00345	0.00195	−1.764	**0.94 (0.89–0.99)**	**0.049 ***
**Gender *(F)***	0.00741	0.03313	0.224	1.42 (0.50–4.07)	0.823
**Length of stay**	0.00589	0.00195	4.704	**4.3 (1.4–15.1)**	**<0.001 ***
**Diabetes Mellitus (*yes)***	0.02280	0.05449	0.418	1.38 (0.32–5.96)	0.676
**Charlson Comorbidity Index**	0.03610	0.01306	2.763	**1.84 (1.24–2.74)**	**0.006 ***
**American Society of Anesthesiology Score**	−0.04104	0.02832	−1.449	0.44 (0.17–1.13)	0.148
**Body Mass Index**	0.00595	0.00425	1.399	1.12 (0.98–1.29)	0.163
**Dural Tears**	0.12622	0.05224	2.416	**4.78 (1.42–16.15)**	**0.016 ***
**Smoking *(yes)***	0.01107	0.03429	0.323	1.28 (0.47–3.49)	0.163
**PLVI**	0.11196	0.07910	1.4415	5.54 (0.51–59.7)	0.158
**M-Score**	0.02145	0.03852	0.557	1.28 (0.39–4.17)	0.578

**Table 3 jcm-12-01387-t003:** Multivariate linear regression of risk factors for PJD. F = female, PLVI = psoas to lumbar vertebral index. Length of stay, age, comorbidity index, ASA score and M score were independent risk factors for infections. OR = odds ratio (95% confidence interval lower–higher). * = significant value.

	Estimate	Se	T	OR (95% CI)	*p* Value
**Age at surgery**	0.00252	0.00146	1.716	**1.42 (1.03–1.96)**	**0.014 ***
**Gender *(F)***	−0.02878	0.02484	−1.159	0.41 (0.09–1.77)	0.248
**Length of stay**	−7.78 × 10^−4^	9.3978 × 10^−4^	−0.828	0.96 (0.86–1.07)	0.408
**Diabetes Mellitus (*yes)***	0.03728	0.04085	0.913	**0.98 (0.93–0.99**	**0.043 ***
**Charlson Comorbidity Index**	−0.00975	0.00979	−0.996	0.73 (0.40–1.34)	0.320
**American Society of** **Anesthesiology Score**	0.01408	0.02124	0.663	1.37 (0.41–4.62)	0.508
**Body Mass Index**	6.9078 × 10^−4^	0.00319	0.216	1.03 (0.86–1.07)	0.829
**Smoking *(yes)***	0.003024	0.02571	1.176	2.80 (0.71–11.06)	0.240
**Dural Tears**	−0.04707	0.03917	−1.201	2.69 (0.38–19.1)	0.230
**PLVI**	0.05418	0.05391	0.913	6.31 (0.24–162.54)	0.362
**M-Score**	−0.03321	0.02888	−1.150	0.33 (0.05–2.23)	0.251

## Data Availability

Not applicable.

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
