# Peer review of "Complications after Posterior Lumbar Fusion for Degenerative Disc Disease: Sarcopenia and Osteopenia as Independent Risk Factors for Infection and Proximal Junctional Disease"

_jcm, 2023, doi:10.3390/jcm12041387_

Round 1
Reviewer 1 Report
I have read the article "Independent risk factors for complications after posterior lumbar fusion: sarcopenia and osteopenia are not related to a higher rate of infections and proximal junctional diseases" with great interest. However, there are some issues that must be addressed to increase the educational value:
Major concerns:
- Were all patients operated on via open technique? or combination including MIS? There is prior literature suggesting differences in infection and junctional outcomes. This could be added to the discussion as well
- There is concern of increased prediction error in the multivariate model. Utilization of forward/backward stepwise regression or other methods is strongly advised
- At p=.049, age is not truly a significant predictor in the proposed model. This should be reassessed and/or at the very least discussed thoroughly
- Statistical assessment of independent predictors in 15 PJD patients is likely severely underpowered. Further power analysis is necessary
- Assessment using validated frailty indices should be added. CCI should not be used as a surrogate alone
- Were there any controls in place for previous or concurrent treatment of sarcopenia or osteopenia? both are regarded as potentially modifiable factors and may be significant confounders
Minor concerns:
- Odds ratios and confidence intervals are not reported in the results
- Figure 2: English translations of the imaging annotations may be helpful to readers
Author Response
Comment 1 Were all patients operated on via open technique? or combination including MIS? There is prior literature suggesting differences in infection and junctional outcomes. This could be added to the discussion as well
Authors’ Response : thank you for the comment. We apologize for not having reported this; they were all operated via open technique. We now specified it in the materials and methods section (lines 67-70).
Comment 2: There is concern of increased prediction error in the multivariate model. Utilization of forward/backward stepwise regression or other methods is strongly advised
Authors’ Response: This is an extremely interesting point. We had the same concern, and, with our statisticians, we thought a lot about the topic. However, the use of stepwise regression in this kind of studies has three major drawbacks: a bias in parameter estimation, inconsistencies among model selection algorithms, and an inappropriate focus or reliance on a single best model, where data are often inadequate to justify confidence. Infact, stepwise regression requires deciding which regression variables should be included in the final analysis and this model selection is conducted through parameter interference (i.e testing whether parameters are significantly different from zero): this can lead to biases in parameters, over-fitting, and incorrect significance tests. Moreover, the algorithm used (forward selection, backward elimination or stepwise), the order of parameter entry (or deletion), and the number of candidate parameters, can allaffect the selected mode. This problem is particularly acute when, as in our study, the predictors are correlated.
[- Akaike, H. (1974) A new look at the statistical model identification. IEEE Transactions on Automatic Control, 19, 716-723.
- Anderson, D.R., Burnham, K.P. & Thompson, W.L. (2000) Null hypothesis testing: problems, prevalence, and an alternative. Journal of Wildlife Management, 64, 912- 923.
- Burnham, K.P. & Anderson, D.R. (2002) Model selection and multimodel inference: a practice information-theoretic approach. Springer Verlag, New York.
- Chatfield, C. (1995) Problem solving: a statistician’s guide. Chapman & Hall, London.]
Comment 3 At p=.049, age is not truly a significant predictor in the proposed model. This should be reassessed and/or at the very least discussed thoroughly
Authors’ Response: you are right, 0.049 represents a “weak” statistical result. We removed age from the discussion and conclusion sections, leaving it only in the tables.
Comment 4:. Statistical assessment of independent predictors in 15 PJD patients is likely severely underpowered. Further power analysis is necessary
Authors’ Response: This is a very important observation, thank you. We added a sentence to the “limits” section regarding the possibility of an underpowered analysis (lines 242-244).
As for the power analysis, we believe it is an indispensable component of planning prospective trial; however, when used to indicate power for retrospective data, it may become meaningless, as the radom component in the study disappears once data are collected. Infact, according to some authors [Zhang Y, Hedo R, Rivera A, Rull R, Richardson S, Tu XM. Post hoc power analysis: is it an informative and meaningful analysis? Gen Psychiatr. 2019 Aug 8;32(4):e100069. doi: 10.1136/gpsych-2019-100069. PMID: 31552383; PMCID: PMC6738696.], if a sample is already selected (as happens in retrospective studies), outcomes are no longer random and power analysis is no longer useful.
Comment 5 Assessment using validated frailty indices should be added. CCI should not be used as a surrogate alone
Authors’ Response: Thank you for the comment, this is an extremely interesting point. The Charlson Comorbidity Index (CCI), created in 1987, has become the most extensively used comorbidity index and is frequently regarded as the gold-standard metric for assessing comorbidity in clinical research. Aim of the present study was to assess the impact of sarcopenia and osteopenia on the outcome of these patients; therefore, we chose to use this validated comorbidity index (and other data such as ASA score and BMI) to be able to adjust the multivariate analysis for potential confounding factors.
Comment 6 Were there any controls in place for previous or concurrent treatment of sarcopenia or osteopenia? both are regarded as potentially modifiable factors and may be significant confounders
Authors’ Response: Thank you for the comment; this is an extremely interesting point. No, there were no controls. We are currently designing a prospective study with the aim of comparing outcomes in patients who are treated for sarcopenia/osteopenia and those who are not.
Comment 7 Odds ratios and confidence intervals are not reported in the results
Authors’ Response: Thank you, you are right. We added all odds ratios and confidence intervals to the tables.
Comment 8 Figure 2: English translations of the imaging annotations may be helpful to readers
Authors’ Response: Thanks for the comment, we deeply apologize. Unfortunately, the imaging annotations are automatically generated by our PACS, it is not possible to change it. We added the translations on the legend figure 2 (AR=Area; Med= Avarage hounsfield unit; DS= standard deviation of the Hounsfield unit; Intervallo= Interval of Hounsfield unit; Raggio=radius).
Reviewer 2 Report
General comments
This study retrospectively examined the independent risk factors for complications after posterior lumbar fusion for degenerative lumbar spine disease. Their results showed that age, comorbidity index, diabetes, dural tear and length stay were independent risk factors for infection and/or proximal junctional disease. This is a well-performed and dedicated study, however, the manuscript has several limitations.
Major concern
Title
Their results showed that sarcopenia and osteopenia were not related to a higher rate of infections and proximal junctional diseases. However, the study has several limitations, including a single-center study, a lack of participants, and a lack of sufficient statistical analysis. Therefore, the authors should avoid adding negative comments to the title.
Materials and Methods
1. Generally, decompression alone or decompression and fusion is the treatment of choice for degenerative lumbar disease. Therefore, indication for fusion surgery should be added in Materials and Methods.
2. The authors should state what the authors adjusted for as potential confounding factors in multivariate linear regression analysis.
Table 1
Baseline characteristic should include the number of fusion levels.
Table 1and Table 3
When discussing PJK or PJF, at least preoperative radiographic parameters are necessary.
Informed Consent Statement
The authors should state how informed consent was obtained from subjects involved in the study.
Author Response
Comment 1: Their results showed that sarcopenia and osteopenia were not related to a higher rate of infections and proximal junctional diseases. However, the study has several limitations, including a single-center study, a lack of participants, and a lack of sufficient statistical analysis. Therefore, the authors should avoid adding negative comments to the title.
Authors’ Response: We completely agree, we changed the title accordingly.
Comment 2: Generally, decompression alone or decompression and fusion is the treatment of choice for degenerative lumbar disease. Therefore, indication for fusion surgery should be added in Materials and Methods.
Authors’ Response: Thank you for the comment, we improved the materials and methods section accordingly
Comment 3: The authors should state what the authors adjusted for as potential confounding factors in multivariate linear regression analysis.
Authors’ Response: You are right, we apologize for not having specified it in the original manuscript. We added it in the materials and methods section (lines 125-127).
Comment 4: Table 1: Baseline characteristic should include the number of fusion levels.
Authors’ Response: We agree, this is an important information. We added it to table 1, along with the results of the univariate analysis.
Comment 5: Table 1and Table 3
When discussing PJK or PJF, at least preoperative radiographic parameters are necessary.
Authors’ Response: Thank you for the comment, this is an extremely important point. When talking about mechanical complications, sagittal balance is important even in short instrumentations; however, our patients were all affected by degenerative disc disease without any case of sagittal or coronal deformity; patients with preoperative sagittal imbalance, flat back or degenerative scoliosis were excluded; infact, all patients underwent short fusions (mean fusion levels 2.84, table 1). We added these considerations to the materials and methods section (lines 69-72) and we deeply apologize for not making it clear in the first version of the manuscript.
Comment 6: Informed Consent Statement
The authors should state how informed consent was obtained from subjects involved in the study.
Author’s Response: Thank you for the comment, we apologize for not making this clear in the first version of the manuscript. The study was approved by our institutional review board (CE-AVEC 208/2022/Oss/IOR) and, before beginning the data collection, all patients were contacted, and informed consent was obtained. We added both these statements in the Materials and Methods section (Lines 66 and 76).
Round 2
Reviewer 1 Report
Thank you to the authors for presenting a revised work of "Complications after posterior lumbar fusion for degenerative disc disease: sarcopenia and osteopenia as independent risk factors for infection and proximal junctional disease".
I still have some concern with the lack of control for potentially modifiable sarcopenia or osteopenia, especially since the conclusion focuses on the lack of its significant impact. This should be added to the discussion and limitations.
I have no other major concerns
Author Response
Comment 1. Thank you to the authors for presenting a revised work of "Complications after posterior lumbar fusion for degenerative disc disease: sarcopenia and osteopenia as independent risk factors for infection and proximal junctional disease". I still have some concern with the lack of control for potentially modifiable sarcopenia or osteopenia, especially since the conclusion focuses on the lack of its significant impact. This should be added to the discussion and limitations. I have no other major concerns
Response to comment 1. Thank you for your comment. We agree: this represents a limit and we added a specific statement to the discussion section. This is the sentence we added: “last but not least, the lack of a control group who received therapy for sarcopenia and/or osteopenia represents a bias and inevitably weakens the conclusions” (lines 244-246)
Reviewer 2 Report
The reviewer really appreciates the author’s response. The manuscript was greatly improved. However, there are still some concerns.
Methods
In this paper, the cases with decompression surgery alone were excluded. The authors should clearly present the indication of fusion surgery for degenerative disc disease in Materials and Methods.
Methods and results
In multivariate linear regression, gender was not included as one of potentially confounding factors. The authors should perform the analysis after adding gender as well as the other confounding factor.
Author Response
Comment 1. In this paper, the cases with decompression surgery alone were excluded. The authors should clearly present the indication of fusion surgery for degenerative disc disease in Materials and Methods.
Thank you for your comment. In the first revision of the manuscript, we specified that patients who underwent decompression alone were excluded; therefore, only patients who underwent fusion were included. We apologize for not specifying our indications for fusion: this is a very controversial topic. In our institution, indications were the following: chronic low back pain with clinical signs of instability; radiographically proven dynamic instability, degenerative spondylolisthesis (>grade 2), central stenosis, significant reduction of disc height, facet degeneration and/or subluxation. We added these indications to Materials and Methods section (lines 70-72)
[Boden SD, Sumner DR, Andersson GBJ, et al: Biologic issues in lumbar spinal fusion. 1995 Focus Issue Meeting on Fusion. Spine 20 (Suppl 24):100S-101S, 1995 7.
Boxall D, Bradford DS, Winter RB, et al: Management of severe spondylolisthesis in children and adolescents. J Bone Joint Surg (Am) 61:479-495, 1979
Bridwell KH, Sedgewick TA, O'Brien MF, et al: The role of fusion and instrumentation in the treatment of degenerative spondylolisthesis with spinal stenosis. J Spinal Disord 6:461-472, 1993]
Comment 2. In multivariate linear regression, gender was not included as one of potentially confounding factors. The authors should perform the analysis after adding gender as well as the other confounding factor.
Thank you for the comment. We apologize: gender was of course considered a confounding factor (along with age, BMI, comorbidity index and ASA), but we forgot to mention it in the Materials and Methods section. We now added it (lines 129-130), thank you for the observation.